# The Reciprocal Association between Fitness Indicators and Sleep Quality in the Context of Recent Sport Injury

**DOI:** 10.3390/ijerph17134810

**Published:** 2020-07-03

**Authors:** Vasileios T. Stavrou, Kyriaki Astara, Zoe Daniil, Konstantinos I. Gourgoulianis, Konstantinos Kalabakas, Dimitrios Karagiannis, George Basdekis

**Affiliations:** 1Laboratory of Cardio-Pulmonary Testing and Pulmonary Rehabilitation, Department of Respiratory Medicine, Faculty of Medicine, University of Thessaly, 41110 Larissa, Greece; kastara@med.uth.gr (K.A.); zdaniil@uth.gr (Z.D.); kgourg@uth.gr (K.I.G.); 2The Medical Project, Prevention, Evaluation and Recovery Center, 41335 Larissa, Greece; kalampakaslarissa@gmail.com (K.K.); dkaragiannis9@gmail.com (D.K.); geobasd@yahoo.gr (G.B.); 3Medical Team, AEL Football Club, 41222 Larissa, Greece

**Keywords:** muscle, injury, oxygen uptake, sleep quality, soccer

## Abstract

The purpose of the study is to investigate whether the oxygen uptake and heart rate at rest, in Greek professional soccer players, are affected by recent injuries, as well as how sleep quality is affected. Forty-two male professional soccer players were included in the study and divided into two groups: injury_group_ (n = 22, age: 21.6 ± 5.4 years, body fat: 11.0 ± 3.9%, total body water: 64.0 ± 2.5%) and no-injury_group_ (n = 20, age: 24.2 ± 5.6 years, body fat: 10.1 ± 2.8%, total body water: 64.3 ± 1.8%). The oxygen uptake at rest (*V*O_2resting_, mL/min/kg) and heart rate (HR, bpm) were recorded in the upright position for 3 min, and the predicted values were calculated. One hour before, the athletes answered the Pittsburgh Sleep Quality Index (PSQI) questionnaire. The results showed a difference between groups (injury_group_ vs. no-injury_group_) in *V*O_2resting_ (7.5 ± 1.4 vs. 5.5 ± 1.2 mL/min/kg, *p* < 0.001) and percent of predicted values (92.5 ± 17.2 vs. 68.3 ± 14.6%, *p* < 0.001) and HR, such as beats per min (100.6 ± 12.8 vs. 93.1 ± 4.6 bpm, *p* = 0.001), percent of predicted values (50.7 ± 6.4 vs. 47.6 ± 2.8%, *p* = 0.003) and sleep quality score (PSQI: 4.9 ± 2.2 vs. 3.1 ± 0.9, *p* = 0.005). Anthropometric characteristics were not different between groups. Oxygen consumption and heart rate at rest are affected by the systemic adaptations due to injury. These pathophysiological changes probably relate to increased blood flow in an attempt to restore the injury area.

## 1. Introduction

Muscle injuries are frequent in soccer and represent about 30% of all injuries and almost 20% at men’s amateur level [1]. Injuries are classified on the basis of their site and their clinical and radiological findings, but the assessment of a precise prognosis remains a crucial point [2]. The criteria of rehabilitation relate to strength tests and self-reported questionnaires. Alterations in the body-loading symmetry are associated with biomechanical conditions and may promote a different contraction for muscles, higher skin temperature in the injury area [3], and probably a higher heart rate.

During exercise, the physiological adaptations regarding to an increase in maximal oxygen uptake include increases in cardiovascular function and skeletal muscle oxidative capacity [4]. Cardiovascular adaptations are related to the type and frequency of exercise, and the kind of sport practiced [5]. According to Godfrey et al. [6], in an inactive period (2–4 weeks), the maximal oxygen uptake declines from 4–20% and blood volume is reduced by 5–12% within the first two days of inactivity in endurance-trained athletes.

The rehabilitation period, as the time period allowed for a return to sports, ranges from 6 to 12 months [7]. A previous sports injury could delay recovery as well as increase the risk for a new one [8]. Additionally, in this period, athletes with injury abstain from training for a long time, thereby reducing the fitness indicators such as oxygen uptake and heart rate variability. Therefore, the presence of a sports injury could hinder the development of athletes’ careers by diminishing their performance. In fact, up to 45% of participants do not return to their preinjury level of sports participation [7]. However, the factors that prevent the reversal of the fitness indicators to preinjury levels after rehabilitation, and insofar as to the degree they extend, remains to be elucidated.

Diminished sleep quality could serve as one of the determinants that possibly undermine recovery. The chronic or acute sleep loss is directly correlated to athletic injuries and a specific disease called “fatigue-related injuries” [9], while, in adolescents, this is related to a greater risk of sports and musculoskeletal injuries [10]. Sleep deprivation increases the risk of overstraining injuries to the locomotor system, while the recent sleep loss also impairs the functional recovery of muscles following injury [9]. Moreover, the pathophysiological pathways implicated probably associate elevated levels of cortisol and inflammation with the decreased levels of testosterone and growth hormone observed during acute and chronic sleep loss that may interfere with tissue repair and growth [9].

The purpose of the study is to investigate the effect of recent injury in oxygen uptake and heart rate at rest among Greek soccer players while examining the effects on sleep quality. We examined possible differences among athletes in quality of sleep due to recent injury. We hypothesize that the end of the competitive period could affect the quality of sleep in both groups, independent of injury [11,12].

## 2. Materials and Methods

### 2.1. Participants

During the preparation period between July 2019 and August 2019, forty-two male professional soccer players from the Greek Super League 1 and 2 were included in the study on a voluntary basis and divided into two groups: Injury_group_ (n = 22) and no-Injury_group_ (n = 20). The muscle injuries (Table 1) was defined as a period of <2 months abstention from regular team training and/or training under a specific rehabilitation program. The no-injury group was defined as a period above 12 months without abstention from competitions and/or training. All volunteers have lived and been trained in Larissa, Greece (<100 m altitude) for at least 24 months. Inclusion criteria were age between 15 to 35 years old, training age >5 years, and participation in Greek Super League 1 and 2 for >18 months. Exclusion criteria were the lack of medical history and recent athletes’ transcripts (<30 days). The study was conducted according to the Helsinki declaration for use in human subjects (N° 58076/14-11-2018, Scientific Council of University Hospital of Larissa, Greece) [13,14]. All the participants, coaches, and parents submitted written consent.

### 2.2. Data Collection

For each athlete, anthropometric and morphological characteristics were recorded (Table 1; body height (Seca 700, Seca Deutschland, Hamburg, Germany), body mass, body mass index, body composition (TanitaMC-980, Tanita Europe BV, Amsterdam, The Netherlands) and percentage of body fat (7 skinfold points measurement, Harpenden, Baty International Ltd, Burgess Hill, UK) [15] and assessment of back and leg flexibility by the Sit and Reach Test) [16]. Moreover, the body surface area was calculated according Mosteller’s [17] formula (BSA_(m^2^)_ = (height_(cm)_ × weight_(kg)_)/3600^½^), the lean body mass was estimated according Boer’s [18] formula (LBM_(kg)_ = 0.407 × weight_(kg)_ + 0.267 × height_(cm)_ − 19.2), both recorded prior to cardiopulmonary function assessment.

The cardiopulmonary function test was performed in upright position for 3 min. The oxygen uptake [*V*O_2(mL/min/kg)_] and heart rate [HR_(bpm)_] in resting (Fitmate MEDCOSMED, Italy) were recorded. Predicted values for oxygen uptake at rest was calculated according Wasserman et al.’s formula [19] (VO_2_ (mL/min/kg) = 150 + (6 × weight_(kg)_)/weight_(kg)_) and maximum heart rate was calculated according the formula (HR_max (bpm)_ = 207 − 0.7 × age) [20].

Prior to the procedures, all athletes answered the Pittsburgh Sleep Quality Index (PSQI) questionnaire [21,22], and it was recorded in their medical history.

All sessions were performed at The Medical Project Center with the environmental temperature at 23 ± 1 °C and humidity of 45 ± 3%. The evaluation was made between 11:00 a.m. to 14:00 p.m. The tests were performed in random sequence for injury and no-injury athletes, and all participants did not have any exercise beforehand.

### 2.3. Statistical Analysis

The Kolmogorov–Smirnov test was used for the normality of the distribution. Mann–Whitney U-test was used between groups, and bivariate correlation analysis was used for statistical comparison between parameters. For each test, the level of significance was set to *p* < 0.05. Continuous variables of interest were characterized by mean values with standard deviation (mean ± SD). The IBM SPSS 21 statistical package (SPSS Inc., Chicago, Illinois, USA) was used for statistical analyses.

## 3. Results

The results did not show any differences between groups in anthropometric and morphological characteristics. In addition, there were differences between groups in back and leg flexibility (*p* = 0.031, Table 1). The results have shown differences between the groups in *V*O_2resting_, such as mL/min/kg as a percent of predicted values (*p* < 0.001, Table 2), and HR_resting_, such as beats per min (*p* < 0.001) and as a percent of predicted heart rate max (*p* = 0.003, Table 2).

The results on sleep quality yielded differences between groups in the variables “*cough or snore loudly*”, “*feel too cold*”, and “*have bad dreams*” (Table 3) and in the scoring of variables (Figure 1). Bivariate correlation analysis showed differences in no-injury_group_ between sleep quality scoring and morphological characteristics (weight, r = 0.471, *p* = 0.027; body surface area, r = 0.452, *p* = 0.035; lean body mass, r = 0.437, *p* = 0.042). Bivariate correlation analysis showed that sleep quality had a moderately positive relationship with O_2_ uptake at rest (r = 0.312, *p* = 0.044). Moreover, results showed positive relationships between variables of the PSQI questionnaire *“cough or snore loudly”* and HR (r = 0.400, *p* = 0.009), and *“have bad dreams”* and HR (r = 0.439, *p* = 0.004) and O_2_ uptake at rest (r = 0.370, *p* = 0.016).

## 4. Discussion

The aim of the study is to investigate whether the oxygen uptake and heart rate at rest are affected by recent injury, and if that recent injury could affect sleep quality. The main findings of our study were that injury_group_ differed in oxygen uptake and heart rate at rest and sleep quality compared to no-injury_group_.

### 4.1. Cardiopulmonary Changes and Injury

Soccer is a complex contact sport with high risks and rates of injury in professional players during practices and matches. Soccer injuries are associated with player age, exercise load, level of play, and standard of training, while adolescent soccer players who are approaching the professional-league level of play are more susceptible to sustaining injuries [23], and this can keep an athlete out of soccer for up to 12 months to repair the injury. All biological tissues require vascular support to survive, and the blood vessels are necessary for repair the injured tissue. The blood vessels lost due to trauma are regenerated, and new tissue formed in response to injury is vascularized [24]. The rate at which blood vessels deliver oxygen, nutrients, growth factors, and circulating cells to repair and regeneration injury tissue has been well documented [24]. *V*O_2max_ is the most important parameter for endurance performance among soccer players [25], as it takes a high level of aerobic fitness to successfully play at a professional level [7]. *V*O_2max_ is a complex process of the proper functioning of lung capacity, cardiac output, and O_2_-carrying capacity of the blood and skeletal muscle efficiency.

The equation of Fick (*V*O_2max_ = (cardiac output × oxygen content of arterial blood) − (cardiac output × oxygen content of mixed venous blood)) interprets the importance of cardiac output as a product of heart rate and stroke volume for O_2_ uptake during exercise, while the cardiac output is an important indicator of how efficiently the heart can meet the body’s demands for perfusion, such as during a high-intensity exercise [26]. The heart and respiratory rates during exercise increase linearly. According to Gąsior et al. [27], the respiratory frequency may considerably modify heart rate and heart rate variability. Our results showed higher values in oxygen uptake and heart rate at rest in athletes with injury, probably due to lower physical activity or the pathophysiological changes relate to increased blood flow that is trying to restore the injured area.

Previous studies observed that a prolonged detraining period could decrease the *V*O_2max_ by 8%, while several weeks of detraining reduced mitochondria in skeletal muscle [28]. In elite soccer players, completion of performance training and the end of the competitive season caused the loss of training-induced physiological and performance adaptations [28]. According to Bogdanis [29], the reactive oxygen species and reactive nitrogen species are produced in skeletal muscles under physiological as well as under pathological conditions, and they consist of the most significant determinants of heart rate variability [30]. Thus, as they are implicated in general fatigue, they, in turn, probably lead to higher values of oxygen uptake and heart rate at rest in injury_group_. Oxygen consumption and heart rate at rest alter the systemic adaptations due to injury, with the cardiac output being reduced after knee surgery in soccer players [31], while the stiffness of the back and leg (42.2 ± 5.0 vs. 48.2 ± 6.1 cm) might relate to lower physical activity due to inactivity. This disorder is associated with sleep loss (r = −0.412, *p* = 0.007) and probably affects the secretion of melatonin, which is responsible for reduced reactive oxygen and nitrogen species, including hydroxyl radical, hydrogen peroxide, singlet oxygen, nitric oxide, and peroxynitrite anion [32,33]. During the abstaining period, systemic adaptations due to inflamed tissues occur. The athlete follows a modified training program without affecting the injured area, altering the body-loading symmetry, and, therefore, the contraction of muscles and skin temperature [3]. In order to avoid disrupting homeostasis in the presence of inflammation, the heart rate fluctuates to activate the cholinergic anti-inflammatory pathway [34]. Increased heart rate activates baroreflex, which results in increased activity of the vagus nerve, which slows the heart rate and restores homeostasis. However, as athletes tend to be in a consistent condition of sympathetic overactivation [35], the heart rate is expected [36], as hypothesized, to be continuously increased, attenuating the vagal anti-inflammatory signals.

### 4.2. Sleep Quality and Injury

Sleep is a key component in an athlete’s recovery process, while reduced sleep may have an impact on injury susceptibility and an increased number and severity of musculoskeletal injuries [22,37]. In high-level athletes, sleep quality is affected by the type of sport and exercise, training frequency and volume, body temperature, altitude, anxiety, psychobiological mechanisms, chronotype, and circadian rhythm [38,39]. Circadian rhythm is the essential driver of homeostasis in the human body and is responsible for the control of systemic physiology through the repetitive fluctuations that affect the physiological and biochemical functions of the organisms and neurogenesis [40]. Our results showed that athletes were classified as good sleepers at the total PSQI score (4.0 ± 2.2) and injury_group_ versus no-injury_group_, respectively (4.9 ± 2.2 vs. 3.1 ± 0.9), compared to previous studies where athletes were classified as poor sleepers when having score >5.5 in PSQI [22]. Although both groups had a relatively low PSQI score, significant differences in specific variables emerged between them, allowing the mapping of a general phenotype of their type of sleep quality.

The loss of physical activity may affect sleep quality, while recent sleep loss impairs the functional recovery of muscles following injury [9]. There is a chronobiological hypothesis for the early mobilization of athletes with injury through physiotherapy and exercise without affecting the injured area in order not to alter the homeostatic mechanisms of chronobiology and core temperature, driven by the circadian rhythm, due to injury. According to Dolezal et al. [41], an interrelationship between sleep and exercise exists. The increased cardiorespiratory fitness was related to improved sleep quality, while daily exercise of moderate-intensity had differential effects on circadian melatonin rhythm and heart rate variability depending on the time of day the exercise was performed due to stimulation of the sympathetic nervous system [41], which improves sleep quality and efficiency rather than duration. Conversely, as our results indicate by the statistically significant positive relationship between sleep quality and fitness indicators, lower O_2_ uptake and HR at rest are associated with poorer sleep quality.

Moreover, our results showed that athletes with injury had higher values in the variables (PSQI) “*cough or snore loudly*”, “*feel too cold*”, and “*have bad dreams*”. The athletes with injury probably presented disturbed sleep due to negative thoughts about the injury and a return to their previous level of sports activity and nervousness about the pain of injury during sleep. This condition is similar to post-traumatic stress disorder (PTSD) and is common among athletes after injury. PTSD patients demonstrate some type of sleep trouble, such as nightmares that are exact replays of the trauma they have experienced, occurring mostly early at night and during different stages of sleep, in contrast to typical dreams [42]. A negative pattern of thinking can interact with inflammatory processes and serve as the foundation of psychosomatic aggravation of an already burdened situation [43,44], namely, musculoskeletal injuries in the sports field.

As an injury forces the athlete to abstain from training, hindering his performance by lowering fitness indicators, sleep disturbances occur. Sleep quality, in turn, instead of enhancing physiotherapy and rehabilitation, ends up undermining muscle recovery. Hence, the systemic disruptions of a recent injury seem to significantly affect sleep quality, and reciprocally, sleep quality hinders rehabilitation.

Future directions could focus on monitoring the effects of a sports injury, as well as designing rehabilitation programs more holistically by taking into account the sleep quality of athletes. Future research could expand on additional aspects that directly or reciprocally affect recovery after sports injuries, in order for athletes to continue their career with their prior-to-injury performance.

## 5. Limitations

Nevertheless, in our study, there were some limitations. The participants were on the same team and possibly had the same behavior. Another limitation is the lack of monitoring of previous injuries (practices and/or matches), chronotype, and cardiorespiratory fitness, so we could not investigate the relationship among types of injuries in different age groups and/or stages of puberty, the intensity of training, and other technical factors.

## 6. Conclusions

Oxygen consumption and heart rate at rest are affected by systemic adaptations due to injury. These pathophysiological changes probably relate to the overactivation of the sympathetic system for the restoration of the injury area. Finally, athletes with injury are advised to continue exercise, with high-intensity resistance training, immediately after surgery, without affecting the injured area. As a result, oxygen supply is enhanced, while the effects of abstaining training are reversed, such as disturbed chronobiology and circadian rhythm as well as muscle damage induced by sleep deprivation. Further research is currently underway.

## Figures and Tables

**Figure 1 ijerph-17-04810-f001:**
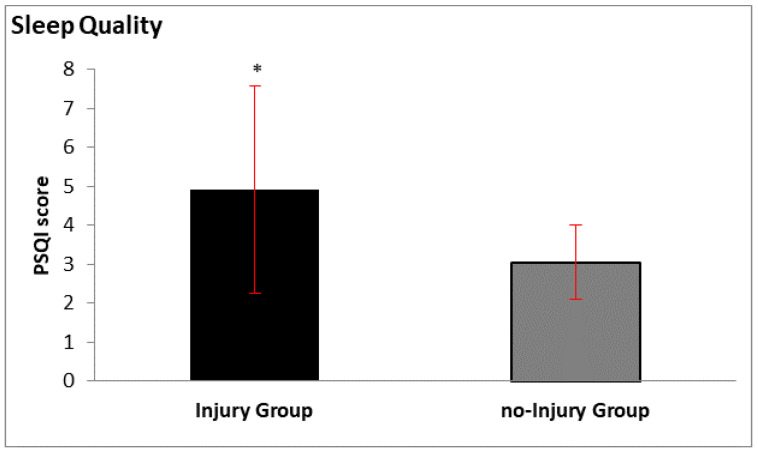
Sleep quality results between groups. * *p* < 0.05.

**Table 1 ijerph-17-04810-t001:** Anthropometric and morphological characteristics between groups. Continuous variables are presented as mean ± standard deviation.

		Injury_group_ (n = 22)	No-Injury_group_ (n = 20)	*p* Value
Age	years	21.6 ± 5.4	24.2 ± 5.6	0.145
Body mass index	kg/m^2^	22.9 ± 1.5	22.9 ± 1.7	0.947
Body surface area	m^2^	1.8 ± 0.2	1.9 ± 0.2	0.667
Lean body mass	kg	58.5 ± 3.8	58.9 ± 3.7	0.753
Body fat	%	11.0 ± 3.9	10.1 ± 2.8	0.415
Total body water	%	64.0 ± 2.5	64.3 ± 1.8	0.712
*Injuries*				
Lateral/medial of thigh muscles	%	27		
Hamstrings muscles	%	36		
Dorsal muscles	%	18		
Abdominal muscles	%	18		
Sit and Reach Test	cm	44.2 ± 5.0	48.2 ± 6.1	0.031

**Table 2 ijerph-17-04810-t002:** Oxygen uptake and heart rate results between groups at rest. Continuous variables are presented as mean ± standard deviation.

		Injury_group_ (n = 22)	No-Injury_group_ (n = 20)	*%* Differences	*p* Value
O_2_ uptake	mL/min/kg	7.5 ± 1.4	5.5 ± 1.2	26.3	<0.001
O_2_ uptake predicted	mL/min/kg	8.1 ± 0.2	8.0 ± 0.2	0.2	0.772
O_2_ uptake	% of predicted	92.5 ± 17.2	68.3 ± 14.6	26.2	<0.001
Heart rate	bpm	100.6 ± 12.8	93.1 ± 4.6	10.6	0.001
Heart rate	% of predicted	50.7 ± 6.4	47.6 ± 2.8	9.3	0.003

**Table 3 ijerph-17-04810-t003:** Pittsburgh Sleep Quality Index (PSQI) results between groups. Continuous variables are presented as mean ± standard deviation.

		Injury_group_ (n = 22)	no-Injury_group_ (n = 20)	*p* Value
**1**	When have you usually gone to bed?	11.4 ± 8.2	11.3 ± 9.3	0.542
**2**	How long has it taken you to fall asleep each night?	14.9 ± 10.7	15.9 ± 8.9	0.748
**3**	What time have you usually gotten up in the morning?	9.3 ± 1.9	9.0 ± 3.3	0.719
**4**	How many hours of actual sleep did you get at night?	7.5 ± 1.2	7.5 ± 1.5	0.954
**5**	Cannot get to sleep within 30 min	0.8 ± 1.0	1.2 ± 1.1	0.322
**6**	Wake up in the middle of the night or early morning	1.2 ± 0.6	0.9 ± 0.4	0.091
**7**	Have to get up to use the bathroom	0.6 ± 0.8	0.5 ± 0.8	0.588
**8**	Cannot breathe comfortably	0.5 ± 0.5	0.2 ± 0.4	0.063
**9**	Cough or snore loudly	0.3 ± 0.5	0.0 ± /	0.018
**10**	Feel too cold	0.0 ± 0.2	0.0 ± /	0.013
**11**	Feel too hot	0.7 ± 1.0	0.4 ± 0.8	0.199
**12**	Have bad dreams	0.7 ± 0.7	0.0 ± /	<0.001
**13**	Have pain	0.0 ± 0.2	0.1 ± 0.2	0.947
**14**	During the past month, how often have you taken medicine to help you sleep?	0.0 ± 0.0	0.1 ± 0.2	0.300
**15**	During the past month, how often have you had trouble staying awake while driving, eating meals, or engaging in social activity?	0.3 ± 0.6	0.2 ± 0.4	0.405
**16**	During the past month, how much of a problem has it been for you to keep up enthusiasm to get things done?	0.2 ± 0.4	0.5 ± 0.8	0.092
**17**	During the past month, how would you rate your sleep quality overall?	0.9 ± 0.7	1.0 ± 0.8	0.717

**Abbreviations:** Questions 5–13 (scale: not during the past month (0), less than once a week (1), once or twice a week (2), three or more times a week (3)). Questions 14–17 (scale: very good (0), fairly good (1), fairly bad (2), very bad (3)).

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
