# Peer review of "The Reciprocal Association between Fitness Indicators and Sleep Quality in the Context of Recent Sport Injury"

_ijerph, 2020, doi:10.3390/ijerph17134810_

Round 1

Reviewer 1 Report

Thank you for your submission to IJERPH. I have decided to limit my comments to major ones since I think they are of more help to the authors. 

Major comments:

  • The article needs to be reread by a native English speaker before submission to another journal in the field. I cannot understand much of the manuscript. 
  • The researchers do not identify a clear gap in the field in which they are trying to exploit and for what reason(s). I do not understand the purpose of measuring resting HR and oxygen consumption following injury or not. How will this advance the field?
  • Title: does this make sense? Is this a question?
  • Introduction: do you identify the gap in the literature? Why are you investigating resting oxygen consumption and HR? How will this add to the field? 
  • Methods: do you have specific details as to injuries the players had? I think this is important and should be noted in the methods. Also, why did you predict oxygen consumption when you had a COSMED device to measure it for you? 
  • Results: was total body hydration measured with the Tanita? I believe this measures total body water and not specifically hydration?
  • Discussion: I do not understand the logic or flow to any of the discussion. I think your main objective here should be ensure the reader understands your results and how they do/don't align with previous. Finally, since there are differences in resting HR and VO2, how does this help us? 
  • Conclusions: I think it is taking it much too far to say that the "pathophysiological changes probably relate to increased blood flow for the restoration of the injury area". How do you know this is the case? You do not have data to support many of these types of comments. 

Minor comments:

Please go back and edit for consistency. For example, use subscripts and superscripts where they should be used and ensure there is a space after every period.

Author Response

Thank you for your submission to IJERPH. I have decided to limit my comments to major ones since I think they are of more help to the authors.

Major comments:

Comment 1: The article needs to be reread by a native English speaker before submission to another journal in the field. I cannot understand much of the manuscript.

Response: We would like to thank the reviewer for his/her helpful comments. We have reshaped the text with better coherence to facilitate the reader's understanding.

Comment 2: The researchers do not identify a clear gap in the field in which they are trying to exploit and for what reason(s). I do not understand the purpose of measuring resting HR and oxygen consumption following injury or not. How will this advance the field?

Response: We would like to thank the reviewer for his/her helpful comments. VO2max and HR at rest have been found to be reduced as athletes abstain due to injury. However, there is scarce evidence concerning the factors that reverse these fitness indicators to pre-injury levels, after rehabilitation, and to what degree they extend. We focused on diminished sleep quality as one of the determinants that could possibly undermine recovery, implying future directions towards monitoring the effects of a sport injury, as well as designing rehabilitation programs. We have added comments in both Introduction and the Discussion section underlying the clarification of the effects of sport injury in recovery, in terms of VO2max/HR - at - rest alterations and sleep quality.

Comment 3: Title: does this make sense? Is this a question?

Response: We would like to thank the reviewer for his/her helpful comments. The title was changed to “The reciprocal association between fitness indicators and sleep quality in the context of recent sport injury.”, as suggested by the reviewer.

Comment 4: Introduction: do you identify the gap in the literature? Why are you investigating resting oxygen consumption and HR? How will this add to the field?

Response: We would like to thank the reviewer for his/her helpful comments. We considered the data that supports the recovery of athletes after injury, in the context of reversing fitness indicators, is limited. To address this matter, we aimed to seek whether differences between injurygroup and no-injurygroup exist. We further searched for a possible association between these differences with affected sleep quality, by hypothesizing that sleep determines the processes of recovery. We acknowledged this gap in the Introduction section, as advised by the reviewer.

Comment 5: Methods: do you have specific details as to injuries the players had? I think this is important and should be noted in the methods. Also, why did you predict oxygen consumption when you had a COSMED device to measure it for you?

Response: We would like to thank the reviewer for his/her helpful comments. Further information concerning sports injuries and oxygen consumption has been added.

Comment 6: Results: was total body hydration measured with the Tanita? I believe this measures total body water and not specifically hydration?

Response: We would like to thank the reviewer for his/her helpful comments. It has been replaced the

“total body hydration” with “total body water”.

Comment 7: Discussion: I do not understand the logic or flow to any of the discussion. I think your main objective here should be ensure the reader understands your results and how they do/don't align with previous. Finally, since there are differences in resting HR and VO2, how does this help us?

Response: We would like to thank the reviewer for his/her helpful comments. The format of the text has been modified for better reader navigation upon the rationale of the study. Furthermore, the reciprocal association of fitness indicators with sleep quality and sport injuries has been supplemented with additional information, based on a more coherent structure of the text.

Comment 8: Conclusions: I think it is taking it much too far to say that the "pathophysiological changes probably relate to increased blood flow for the restoration of the injury area". How do you know this is the case? You do not have data to support many of these types of comments.

Response: We would like to thank the reviewer for his/her helpful comments. This conclusion was extracted by the physiologic effects of heart rate variability, that occurs during several autoinflammatory musculoskeletal diseases and injuries. In order to restore homeostasis, heart rate variability enhances a neuronal circuit. Increased heart rate activates a reflex which results in to increased activity of the vagus nerve, which, in turn, slows heart rate and restores homeostasis. As anti - inflammatory processes are impaired due to the antagonizing effects of sympathetic system, the inflamed tissues fail to recover properly. Although blood flow in the injured region is altered, we rephrased the conclusion to “These pathophysiological changes probably relate to the overactivation of the sympathetic system for the restoration of the injury area” to focus on the broader correlation between HR and sport injury. We added relevant data in the Discussion section, to support our assumptions, as suggested by the reviewer.

Minor comments:

Please go back and edit for consistency. For example, use subscripts and superscripts where they should be used and ensure there is a space after every period.

Minor comments:

Comment 7: Please go back and edit for consistency. For example, use subscripts and superscripts where they should be used and ensure there is a space after every period.

Response: We once again would like to thank the reviewer for the careful review of the manuscript. We have reviewed the paper and have proceeded with the corrections as correctly requested.

Reviewer 2 Report

This is an interesting aim with the quality of life scope. 

This is a well-written manuscript with an important clinical message, and should be of great interest to the readers of International Journal of Environmental Research and Public Health. However, from my point of view, authors should include the following requirements:

First of all, I suggest change the section title, reflecting methods and size, due to the fact, I believe is better include some relative to a cross-sectional observational study, because in my opinion it reflects better the goal of the study.

The redaction is clear and concise with appropriated scientific terms. Evaluating quality of sleep in recent sport injuries: is a trend topic in the current research literature and may be a main focus of interest for readers.

The sample size calculation, structured tables and methodology are adequate and provide important contents.

Therefore, this study may support considerations about the sport injuries affects quality of sleep.

On the other hand, Introduction section may be improved by adding new information in order to provide an adequate state-of-the-art including some references. I suggest to include this reference to complete this requirement relative to thermography and soccer injuries:

Lines 25-35 Page 1

Rodriguez-Sanz, D.; Losa-Iglesias, M.E.; Becerro de Bengoa-Vallejo, R.; Palomo-Lopez, P.; Beltran-Alacreu, H.; Calvo-Lobo, C.; Navarro-Flores, E.; Lopez-Lopez, D. Skin temperature in youth soccer players with functional equinus and non-equinus condition after running. J. Eur. Acad. Dermatology Venereol. 2018, 32, 2020–2024.

Methods are well-designed with relevant and complete information. Correct sample size calculations, good description of the properties of the outcome measurements as well as detailed statistical analyses were included. Tables, figures and redaction of the results are presented in a correct way providing a good presentation of the main finding of the study.

In line 55 page 2 in methods section I suggest authors must include a reference to Ethics requirements Helsinki declaration and Strobe methods:

-Vandenbroucke, J.P.; von Elm, E.; Altman, D.G.; Gøtzsche, P.C.; Mulrow, C.D.; Pocock, S.J.; Poole, C.; Schlesselman, J.J.; Egger, M.; STROBE Initiative Strengthening the Reporting of Observational Studies in Epidemiology (STROBE): explanation and elaboration. Int. J. Surg. 2014, 12, 1500–24.

-Holt, G.R. Declaration of Helsinki—The World’s Document of Conscience and Responsibility. South. Med. J. 2014, 107, 407–407.

Discussion section may include future research studies secondary to the current findings of this study. Clinical considerations, limitations and overall discussion are well-presented, but future research may be useful in order to propose future research regarding this field. I suggest including comparing with another instrumental technique for example eye tracking:

-Guan, Z.; Cutrell, E. (2007). “An Eye Tracking Study of the Effect of Target Rank on Web Search.” CHI 2007, April 28–May 3, 2007, San Jose, California, USA.

Author Response

This is an interesting aim with the quality of life scope. This is a well-written manuscript with an important clinical message, and should be of great interest to the readers of International Journal of Environmental Research and Public Health. However, from my point of view, authors should include the following requirements:

Comment 1: First of all, I suggest change the section title, reflecting methods and size, due to the fact, I believe is better include some relative to a cross-sectional observational study, because in my opinion it reflects better the goal of the study.

Response: We would like to thank the reviewer for his/her helpful comments. The title was changed to “The reciprocal association between fitness indicators and sleep quality in the context of recent sport injury.”, as suggested by the reviewer.

The redaction is clear and concise with appropriated scientific terms. Evaluating quality of sleep in recent sport injuries: is a trend topic in the current research literature and may be a main focus of interest for readers. The sample size calculation, structured tables and methodology are adequate and provide important contents. Therefore, this study may support considerations about the sport injuries affects quality of sleep.

Comment 2: On the other hand, Introduction section may be improved by adding new information in order to provide an adequate state-of-the-art including some references. I suggest to include this reference to complete this requirement relative to thermography and soccer injuries:

Lines 25-35 Page 1 (Rodriguez-Sanz, D.; Losa-Iglesias, M.E.; Becerro de Bengoa-Vallejo, R.; Palomo-Lopez, P.; Beltran-Alacreu, H.; Calvo-Lobo, C.; Navarro-Flores, E.; Lopez-Lopez, D. Skin temperature in youth soccer players with functional equinus and non-equinus condition after running. J. Eur. Acad. Dermatology Venereol. 2018, 32, 2020–2024.)

Response: We would like to thank the reviewer for his/her helpful comments. We used this very helpfully reference. It has been added more information in Introduction.

Methods are well-designed with relevant and complete information. Correct sample size calculations, good description of the properties of the outcome measurements as well as detailed statistical analyses were included. Tables, figures and redaction of the results are presented in a correct way providing a good presentation of the main finding of the study.

Comment 3: In line 55 page 2 in methods section I suggest authors must include a reference to Ethics requirements Helsinki declaration and Strobe methods: -Vandenbroucke, J.P.; von Elm, E.; Altman, D.G.; Gøtzsche, P.C.; Mulrow, C.D.; Pocock, S.J.; Poole, C.; Schlesselman, J.J.; Egger, M.; STROBE Initiative Strengthening the Reporting of Observational Studies in Epidemiology (STROBE): explanation and elaboration. Int. J. Surg. 2014, 12, 1500–24. -Holt, G.R. Declaration of Helsinki—The World’s Document of Conscience and Responsibility. South. Med. J. 2014, 107, 407–407.

Response: We would like to thank the reviewer for his/her helpful comments. We have included the references in Helsinki declaration, as suggested by the reviewer.

Comment 4: Discussion section may include future research studies secondary to the current findings of this study. Clinical considerations, limitations and overall discussion are well-presented, but future research may be useful in order to propose future research regarding this field. I suggest including comparing with another instrumental technique for example eye tracking:  -Guan, Z.; Cutrell, E. (2007). “An Eye Tracking Study of the Effect of Target Rank on Web Search.” CHI 2007, April 28–May 3, 2007, San Jose, California, USA.

Response: We would like to thank the reviewer for his/her helpful comments. The Discussion section was supplemented with comment regarding future directions and researches, as proposed by the reviewer.

Reviewer 3 Report

Dear authors:
It has been a pleasure to review your paper about “The recent sport injury can affect the quality of sleep?” but I have observed some of errors that it’s necessary to change it You can see below the recommendation

Title: It should include the type of study

Introduction: It is very brief , I understand that it is a brief paper but you need to improve and justify the information, you only have two refences

In section method

This section is very poor and brief, it’s impossible understand the paper

I can’t find the exclusion criteria. Please can you include them?

How did you calculate the sample size, can you include this in the text?

Can you increase the data collect section? it’s impossible understand the paper with this information

What criteria did you use to determine the injury or not of the player??? What consensus????

The PSQI is validate in Greek???

Result:

Can you write the p values complete not NS?

Discussion

Please can you include clinical implications in the discussion section?

Author Response

Dear authors:

It has been a pleasure to review your paper about “The recent sport injury can affect the quality of sleep?” but I have observed some of errors that it’s necessary to change it You can see below the recommendation

Comment 1: Title: It should include the type of study

Response: We would like to thank the reviewer for his/her helpful comments. The title was changed to “The reciprocal association between fitness indicators and sleep quality in the context of recent sport injury.”, as suggested by the reviewer.

Comment 2: Introduction: It is very brief, I understand that it is a brief paper but you need to improve and justify the information, you only have two refences

Response: We would like to thank the reviewer for his/her helpful comments. We have added more information and references in Introduction section, as proposed by the reviewer.

In section method

This section is very poor and brief, it’s impossible understand the paper

Comment 3: I can’t find the exclusion criteria. Please can you include them?

Response: We would like to thank the reviewer for his/her helpful comments. We have added exclusion and inclusion criteria in Methods section.

Comment 4: How did you calculate the sample size, can you include this in the text?

Response: We would like to thank the reviewer for his/her helpful comments. We have added "During preparation period between July 2019 and August 2019, forty-two male professional ..."in the Method section.

Comment 5: Can you increase the data collect section? it’s impossible understand the paper with this information

Response: We would like to thank the reviewer for his/her helpful comments. We have rewrite the "data collect" section, as proposed by the reviewer.

non-participation in competitions

Comment 6: What criteria did you use to determine the injury or not of the player??? What consensus????

Response: We would like to thank the reviewer for his/her helpful comments. We have added clarifications about the criteria of injury group in Methods section.

Comment 7: The PSQI is validate in Greek???

Response: We would like to thank the reviewer for his/her helpful comments. The PSQI is validation in Greek and we have added corresponding reference.

Result:

Comment 9: Can you write the p values complete not NS?

Response: We would like to thank the reviewer for his/her helpful comments. The p values have been added as requested in the whole text and tables as well as in Abstract. Discussion

Comment 10: Please can you include clinical implications in the discussion section?

Response: We would like to thank the reviewer for his/her helpful comments. We have added clinical implications in the discussion section.

Reviewer 4 Report

Abstract

Line 11 - Please consider changing “is affected by the recent injury” for “is affected by a recent injury

Introduction

Line 43 and 44 – the hypothesis did not match with the purpose of the study. Do you have two aims, or your focus is sleep quality? Please clarify.

Methods

The injury group were training at the moment of the data collection or were they still in rehabilitation?

Discussion

Line 116 to 120 – please provide a reference

Line 130 - due to injury or to the inactivity, thus detraining?

Line 132 – the authors stated “may be associated”, why not include a correlation or effect size?

Line 144 and 145 – Thus, these players were not training?

Line 163 – “motor pattern” or behavior?

Please clarify the "injury group status", i.e., if they were active or not and what is the average time from the moment of the injury and the present study.

Author Response

Abstract

Comment 1: Line 11 - Please consider changing “is affected by the recent injury” for “is affected by a recent injury

Response: We would like to thank the reviewer for his/her helpful comments. We have replaced “is affected by the recent injury” with “is affected by a recent injury”.

Introduction

Comment 2: Line 43 and 44 – the hypothesis did not match with the purpose of the study. Do you have two aims, or your focus is sleep quality? Please clarify.

Response: We would like to thank the reviewer for his/her helpful comments. We have replaced and clarified the aim of our study “The purpose of the study was to investigate the effect of recent injury in oxygen uptake and heart rate in resting among in Greek soccer players, while examining the effects on sleep quality. We examined possible differences among athletes in quality of sleep due to recent injury. We hypothesized that the end of the competitive period could affect the quality of sleep in both groups independent of injury 11,12.” as proposed by the reviewer.

Methods

Comment 3: The injury group were training at the moment of the data collection or were they still in rehabilitation?

Response: We would like to thank the reviewer for his/her helpful comments. We have added clarification about the injurygroup in Methods section.

Discussion

Comment 4: Line 116 to 120 – please provide a reference

Response: We would like to thank the reviewer for his/her helpful comments. We have added reference.

Comment 5: Line 130 - due to injury or to the inactivity, thus detraining?

Response: We would like to thank the reviewer for his/her helpful comments. We have replaced “due to the injury” with “due to the inactivity”.

Comment 6 Line 132 – the authors stated “may be associated”, why not include a correlation or effect size?

Response: We would like to thank the reviewer for his/her helpful comments. We have added “This disorder associated with sleep loss (r=-.412, p=0.007)……”.

Comment 7: Line 144 and 145 – Thus, these players were not training?

Response: We would like to thank the reviewer for his/her helpful comments. We have added more information in Methods section, about the athletes

Comment 8: Line 163 – “motor pattern” or behavior?

Response: We would like to thank the reviewer for his/her helpful comments. We have replaced “motor pattern” with “behavior”.

Comment 9: Please clarify the "injury group status", i.e., if they were active or not and what is the average time from the moment of the injury and the present study.

Response: We would like to thank the reviewer for his/her helpful comments. We have added clarification about the criteria of injurygroup in Methods section.

Round 2

Reviewer 1 Report

Major comments:

  • The article needs to be reread by a native English speaker before submission to another journal in the field. I still cannot understand much of the manuscript. For example, the first sentence of the abstract "the oxygen uptake and heart rate in resting" - this does not make sense and right away puts the manuscript in a negative light. 
  • For such a prestigious journal there is a lack of purpose, significance, and scientific soundness for this study. In my opinion it does not add a great deal to the practical or scientific field.
  • Many of my comments have not been addressed across the entire manuscript. Again, let's stay with the abstract - "total body hydration" remains in the abstract. You did not collect data to measure one's total body hydration rather total body water.
  • Many comments relate to probable cause for reasoning of data while the researchers did not collect variables specific to this probable cause. There is lack of clarity between results and discussion/conclusions. Again, you mention "probably relate to the overactivation of the sympathetic system for the restoration of the injury area" - you did not measure anything specifically related to the sympathetic nervous (i.e., in my opinion resting HR is too long of a stretch to be specifically related to the SNS) to nor do you explain this comment in way/shape/form. 

Reviewer 3 Report

Dear authors:
It has been a pleasure to review your paper  “The reciprocal association between fitness indicators and sleep quality in the context of recent sport injury”  again. The paper can be accepted in this form, I only suggested

Title: It should include the type of study. Can you include cross-sectional study